# Hydrogen Sulfide Adsorption from Natural Gas Using Silver-Modified 13X Molecular Sieve

**DOI:** 10.3390/ma17010165

**Published:** 2023-12-28

**Authors:** Mirzokhid Abdirakhimov, Mohsen H. Al-Rashed, Janusz Wójcik

**Affiliations:** 1Department of Chemical Engineering and Process Design, Silesian University of Technology, 44-100 Gliwice, Poland; mirzokhid.abdirakhimov@polsl.pl; 2Public Authority for Applied Education & Training, Department of Chemical Engineering, College of Technological Studies, Kuwait City 70654, Kuwait; mhj.alrashed@paaet.edu.kw

**Keywords:** H_2_S adsorption, 13X molecular sieve, adsorption capacity, ion exchange, silver modification

## Abstract

The removal of hydrogen sulfide from natural gas and other gases such as biogas, refinery gases, and coal gas is required because it is toxic and corrosive, even in traces. Zeolites are widely used in the removal of H_2_S from the abovementioned gases. In this work, we prepared an Ag-exchanged 13X molecular sieve by using different concentrations of AgNO_3_ to increase its adsorption properties. XRD, SEM, and BET techniques were used to characterize samples. To determine the adsorption properties of each of the samples, a laboratory setup with a fixed-bed adsorber was utilized. The adsorption capacity of modified 13X increased when the molar concentration of AgNO_3_ increased from 0.02 M to 0.05 M. However, the breakthrough time was attained quicker at a high molar concentration of 0.1 M AgNO_3_, indicating a low adsorption capacity. When compared to unmodified 13X, the adsorption capacity of AgII-13X increased by about 50 times. The results of this study suggest that the silver-modified 13X molecular sieve is highly effective at extracting H_2_S from natural gas.

## 1. Introduction

When compared to other fuels, natural gas produces fewer greenhouse gases during combustion. According to the International Energy Agency (IEA) [1], it accounts for more than one-third of global energy consumption and is expected to expand substantially in all scenarios. Although natural gas is considered a pure fuel, it contains carbon dioxide (CO_2_), hydrogen sulfide (H_2_S), and other sulfur compounds such as mercaptans (R-SH), carbonyl sulfide (COS), and carbon disulfide (CS_2_) that need to be removed. The term “sour” natural gas refers to natural gas that contains hydrogen sulfide [2]. Because H_2_S is toxic and corrosive in nature, even a trace amount of it should be removed or reduced substantially. To meet the natural gas sales specification, the H_2_S concentration must be less than 4 ppmv [3]. The procedure of “sweetening” of natural gas entails the extraction of acid gases, with a primary focus on hydrogen sulfide. There exist four distinct methodologies employed for the removal of H_2_S, including absorption, adsorption, membrane separation, and cryogenic distillation. Absorption and adsorption are widely recognized processes employed in the natural gas sector for the removal of acid gases. Absorption is mainly based on the simultaneous selective bulk absorption of H_2_S and CO_2_ acid gases using various solvents [4]. Alkanolamine solvents and their blends are extensively utilized as solvent [5,6]. The process is characterized by high energy consumption, as well as issues related to equipment corrosion and solvent loss. Adsorption is the process by which molecules adhere to the adsorbent’s surface. In this process, activated carbon [7,8,9], MOFs [10,11], metal oxides [12,13], and zeolites [14,15] are commonly used to eliminate H_2_S from various gases. The adsorbents described above each have their own set of drawbacks, which prevents them from being utilized in industrial settings. In the case of activated carbon, for instance, it possesses a high adsorption capacity and is superior to other adsorbents; yet its regeneration process is very challenging. Metal oxides exhibit a high affinity for H_2_S, but they have a poor surface area and a lack of pores [16]. Furthermore, they are able to demonstrate their activity at high temperatures, which leads to increased costs related to energy consumption and obstacles in the process. However, zeolites are preferable to other adsorbents for their characteristics, such as high temperature stability, large surface area, regenerability, and low cost. They are crystalline aluminosilicates of alkaline and alkaline earth metals. Their open three-dimensional framework structures are made of corner-sharing AlO_4_ and SiO_4_ tetrahedra. Adsorbents such as Linde Type A (4A, 5A) and Faujasite (13X) molecular sieves are widely used in industry to remove acid gases. The use of 13X Faujasite zeolite for the extraction of H_2_S from different mixtures has shown promising results [17,18,19,20]. Although there are numerous approaches in H_2_S capture, choosing the best depends on factors including economics and sustainability: a sorbent having strong adsorption capabilities may not be favored due to its expense. For example, MOFs display good adsorption performance towards H_2_S, but they tend to be pricier than zeolites and amines [21]. Appropriate H_2_S removal techniques can be chosen based on process parameters and operating requirements.

As a result of the increased demand for natural gas, the sour components contained in it will have to be reduced appreciably. This requires the synthesis of new adsorbents or improvements in the adsorption properties of the existing ones. The introduction of various metals, such as Cu, Zn, Co, and Ag, into zeolites has been extensively studied to increase the adsorption performance toward H_2_S. Barelli et al. [22] conducted a study on the use of Cu-exchanged 13X for the removal of H_2_S from biogas and found that it exhibits high adsorption capacity in a wide range of operating conditions. Nguyen and colleagues [23] also hydrothermally produced Faujasite zeolite and ion-traded it with divalent metal ions (Co^2+^, Mn^2+^, Ni^2+^, Cu^2+^, Ca^2+^, and Zn^2+^) to study H_2_S adsorption. When compared to non-modified Faujasite X, the adsorption capacity of CoX, CuX, and ZnX zeolites was more than 7, 13, and 24 times higher, respectively. Chen et al. [24] studied the adsorption performance of AgX, CoX, and ZnX zeolite synthesized by ion-exchanging of X zeolite for Claus tail gas desulfurization. The authors reported that AgX had a high adsorption capacity for H_2_S and COS in comparison to other samples. Kumar et al. [25] investigated Ag and Cu-modified X and Y Faujasite to remove H_2_S from gas streams containing He, N_2_, CO_2_, CO, and H_2_O. Aqueous solutions of 0.05 M AgNO_3_ and 0.5 M Cu(NO_3_)∙3H_2_O were used as a source of Ag and Cu. The experiments were carried out at both room temperature and 150 °C by simulating a range of H_2_S, CO_2_, CO, He, N_2,_ and H_2_O gases. AgX and AgY were able to adsorb H_2_S despite the presence of other gases while CuX and CuY failed in the presence of 2% CO. It was found that Ag-exchanged Faujasite had strong selectivity towards H_2_S with adsorption capacity of 1.8 mmol/g whereas Cu-exchanged Faujasite was susceptible to CO adsorption. Kulawong et al. [26] examined Ag-exchanged NaX zeolite as a means of removing H_2_S from an anaerobic digestor reactor. The author’s findings revealed that an increase in the loading of Ag positively impacted the adsorption of H_2_S. Zhu et al. [27] used an impregnation approach to modify 13X zeolite with Ag and Cu for the removal of organic sulfur compounds from nitrogen. Metal ion concentrations ranging from 0.01 to 1.0 mol/L were investigated, with a final value of 0.1 mol/L chosen for further investigations. The saturated adsorption capacity of Ag-impregnated 13X zeolite was 64.8 mg S/g. The effective adsorption capacity, however, was not calculated. There is still a lack of clarity on whether or not the silver-impregnated 13X zeolite exhibits such adsorption capacity toward H_2_S rather than organic sulfur compounds.

As mentioned earlier, H_2_S is mainly found with methane in various gases including biogas, natural gas, refinery gas, coal gas, and other gases. Therefore, in order to assess adsorbents in a real-world setting, it is crucial to conduct tests using real gas mixtures. The aforementioned research attempts utilized a diverse range of synthetic gases instead of natural gas. To our knowledge, there has been an absence of study pertaining to the adsorption of H_2_S from methane.

In this work, we conducted experiments with a real natural gas mixture to study the effect of Ag-modified 13X molecular sieves on the removal of H_2_S. Ag-modified 13X samples were prepared by the ion-exchange method. In addition, we investigated the effect of the inlet H_2_S concentrations and exchange rates of the Ag ions on the adsorption operation. The current study showed significant efficacy in the development of a silver-exchanged 13X molecular sieve utilized for the extraction of H_2_S from natural gas.

## 2. Materials and Methods

### 2.1. Reagents and Materials

The substrate materials were a conventionally available 13X molecular sieve (from Hurtland LLC, Poland), AgNO_3_ (from Stanlab LLC, Lublin, Poland), methane 2.5 (Siad Poland LLC, Ruda Śląska, Poland), 5000 ppm H_2_S in CH_4_ (from Air Liquide Polska LLC, Kraków, Poland), and deionized water.

### 2.2. Synthesis of Ag Ion-Exchanged 13X

Ag ion-exchanged 13X was prepared by stirring 10 g of 13X molecular sieve in various molar concentrations of AgNO_3_ water solution (0.02 M, 0.05 M, and 0.1 M in 200 mL) for 24 h. Ag ion-exchanged 13X molecular sieves were labelled as AgI-13X, AgII-13X, and AgIII-13X, respectively. Then, the samples were washed with deionized water, filtered, and dried at 110 °C for 12 h. Calcination was carried out at 600 °C overnight in the oven. Samples were cooled and kept in a desiccator.

### 2.3. Characterization

The phase composition of samples was determined using a powder X-ray diffractometer (Seifert 3003TT) with a Cu X-ray tube (kλ1 = 1.540598 Å, kλ2 = 1.544426 Å, kβ = 139,225 Å). The powder samples were analyzed between 5° and 80° of 2Theta with 0.05° step. In order to validate the crystal structures, the X-ray diffraction patterns that were acquired were compared with the information that was collected from the Joint Committee on Powder Diffraction Standards (JCPDS) [28]. Morphological features of the sample surfaces were obtained by scanning electron microscope (SEM) images using a Phenom ProX SEM (Phenom-World BV, Eindhoven, The Netherlands). For the SEM imaging, the samples were coated with a thin layer of gold and mounted on a slab using double-sided tape. The elemental analysis of the samples was also carried out by energy dispersive X-ray spectroscopy (EDS) during SEM image acquisition (Phenom-World BV, Eindhoven, The Netherlands). The BET surface of the samples was measured using a Micromeritics ASAP 2020 adsorption analyzer (Micromeritics Inc., Norcross, GA, USA).

### 2.4. H_2_S Gas Separation

A laboratory scale setup was used to carry out H_2_S adsorption, as shown in Figure 1. Ten grams of adsorbent was placed in the Teflon adsorber (40 mm long and 1.5 mm internal diameter) and attached to the system. The primary objective of employing 10 g of adsorbent was to replicate the authentic process, as the flow rate of 400 mL/min may prevent H_2_S molecules present in methane coming into contact with the surface of the adsorbent. As a result, analyzers in the outlet flow may quickly detect the presence of H_2_S gas that is not in direct contact with the adsorbent, allowing for the early detection of breakthrough time. The installation was carefully checked to ensure all connections had no leakage. The mixture was fed directly to a scrubber unit until the desired H_2_S concentration in methane was achieved. Desired concentrations (within the 150–500 ppm range) of H_2_S were introduced at the top of the adsorption column with a flow rate of 400 mL/min under atmospheric pressure. A rotameter was used to keep the mixed gas flow rate constant at 400 mL/min for each of the experiments. To measure initial and breakthrough H_2_S concentrations, two analyzers (Southland Sensing Ltd., Ontario, CA, USA) were installed before and after the adsorber. Analyzers are able to measure the concentration of hydrogen sulfide in a broad range, from 0 to 2000 ppm. H_2_S concentrations were recorded in the input and output every second to obtain accurate results. Outlet gas was treated with a NaOH (pH 13.4) solution and burned before being released. Phenolphthalein was used to indicate the H_2_S saturation of the NaOH solution. Pipelines and fittings made of stainless steel were used to prevent corrosion.

### 2.5. Adsorption Capacity

Adsorption capacity is defined as the ratio of adsorbed molecular amount to adsorbent mass, and it is typically represented in units of mmol/g or mg/g [15]. The efficiency of the adsorbent is assessed by finding its adsorption capacity. There are two types of adsorption capacity, effective and saturated adsorption capacity. The former is calculated when the outlet concentration of H_2_S is 1 ppm, regardless of what the inlet concentration is. The latter is calculated when the outlet H_2_S concentration reaches the initial concentration. Consequently, the saturated adsorption capacity is always greater than the effective adsorption capacity. Since, in most cases, effective adsorption capacity is important, in this study we were limited to its calculation alone. The following equation was used to calculate effective adsorption capacity.
(1)$$C_{ads}=\frac{Q_{tot}\cdot MW\cdot \left[C_{in}\cdot t_{1}-(t_{1}-t_{0})\cdot 0.5\right]}{V_{m}\cdot m\cdot 10^{3}}$$
where *Q_tot_ =* total gas flow rate (Nl/h);

*MW* = molecular weight of H_2_S (g/mol);*C_in_* = inlet H_2_S concentration (ppmv);*t*_1_ = breakthrough time when the outlet concentration is 1 ppmv (h);*t*_0_ = breakthrough time at the last detection of 0 ppmv (h);V_m_ = molar volume (24,414 Nl/mol);*M* = mass of adsorbent material (g).

### 2.6. Methodology

To evaluate the material’s efficacy in removing H_2_S, dynamic tests were conducted. We used 13X spherical pellets with an average particle diameter in the 3–5 mm range and Ag ion-exchanged samples were used for the main part of the experiments. The zeolite adsorbents were heated in an oven at 110 °C overnight in order to remove any residual gases and traces of humidity that were present inside the pores. After the heating process was complete, the zeolite adsorbents were cooled and stored in a desiccator. The amount of samples was 10.00 g for each test measured after thermal treatment. Adsorption runs were carried out on zeolite samples to obtain for each set of operating conditions the corresponding breakthrough curve. To produce an adequate concentration of H_2_S (i.e., 150 ppm, 300 pm, 500 ppm), 5000 ppm of H_2_S in methane was diluted with pure methane. When the inlet analyzer showed desired concentration of H_2_S, the generated gas was allowed to flow to the H_2_S scrubber within a certain amount of time, in order to guarantee that the proper concentration of H_2_S was reached. Afterwards, a gas mixture containing the desired concentration of H_2_S was passed through the adsorber and inlet and outlet H_2_S concentrations were measured at every second to achieve accurate results.

## 3. Results and discussion

### 3.1. XRD Analysis

The XRD patterns corresponding to 13X, AgI-13X, AgII-13X and AgIII-13X are presented in Figure 2. The samples feature significant crystallinity, as shown by the strength and broadening of the XRD peaks. The investigated samples showed mainly a crystalline phase composed of sodium aluminum silicate (Na_14_Al_14_Si_34_O_96_) according to the PDF card no. 04-010-5065. The main diffraction peaks at 2 θ = 6.1, 10.0, 11.9, 15.2, 18.3, 20.1l, 23.2, 26.9, 31.0 are characteristic of the Faujasite structure (JCPDS No: 12-0228). Between 5° and 15° of 2Theta, the characteristic bump was identified, characteristic of the amorphous phase. In this part, some of the Ag-based compounds could be identified; however, not in the crystalline phase. No significant difference was observed between fresh 13X and the Ag-modified samples. It should be noted, however, that the intensity of some peaks for the samples that had been exchanged with silver was diminished. The distinction might be particularly apparent in the X-ray diffraction (XRD) pattern of AgIII-13X. The absence of measured values for 10°, 16°, 29°, 35°, and 42° of Theta2 indicates that there was minimal lattice collapse following the process of ion-exchanging. The results suggest that the crystal structure of the 13X molecular sieve stayed unchanged following Ag ion-exchange treatment. Furthermore, the absence of any discernible peaks on the spectra implies that no Ag_2_O was generated on the 13X molecular sieve after silver modification.

### 3.2. SEM Images

Figure 3 shows the SEM images of the samples. The framework of the Faujasite zeolite was built by connecting sodalite cages by six rings [29]. SEM analysis verified that both silver-modified and fresh 13X were composed of very well-shaped crystallites with a spherical (octahedral) morphology. However, the fresh 13X molecular sieve showed much smoother surfaces as compared to the silver ion-exchanged 13X particles. Moreover, after the ion-exchange process, some of 13X particles cracked, suggesting that lattice destruction might happen during the ion exchange or high-temperature calcination. These findings are compatible with the findings of the XRD investigation. Additionally, it is evident that when the ion exchange rate rose in silver-modified 13X, the quantity of minor additives in its surface morphology also increased. It is possible to attribute the smaller particles found between the zeolite crystals to the binder, which was composed of clay and was utilized in the process of shaping the crystals into beads.

Compositional characterization of the samples was achieved by energy-dispersive X-ray spectroscopy (EDS) during SEM image acquisition. The element content in the samples is shown in Table 1. It can be seen that the high-weight concentration of silver in AgI-13X (10.24 wt. %), AgII-13X (21.38 wt. %), and AgIII-13X (32.38 wt. %) confirmed ion exchange was carried out successfully. Since alkaline metals tend to exchange with silver ions, it can be confirmed that Na^+^ cations contained in zeolite were replaced by Ag^+^ cations. This can be confirmed by a decrease in Na^+^ cations and an increase in Ag^+^ cations (Table 1). Knowing that Na+ can be fully replaced by Ag+ ion [24] and having confirmed the existence of Na+ in the silver-modified samples, we can conclude that no Ag_2_O was produced. Additionally, the findings were verified by XRD analysis.

### 3.3. BET Analysis 

The specific surface area was determined by low-temperature nitrogen adsorption (ASAP 2020, Micromeritics Inc., Norcross, GA, USA) using the Brunauer–Emmett–Teller equation [30]. Prior to taking the nitrogen adsorption measurements, each selected sample was outgassed for 24 h at 350 °C. The BET surface area of the samples is given in Table 2. The BET surface area was calculated as 501.33 m^2^/g for 13X. However, BET surface area was influenced by the ion exchange of Ag. An increase in the quantity of silver ions most likely contributed to a decrease in the BET surface area, since the silver ion exchange of 13X resulted in a marginal decrease in the specific surface area of AgIII-13X from 501.33 to 405 m^2^/g. Similar findings were given by Chen et al., who found the BET-specific surface area of silver-exchanged X was reduced by 33% [24].

### 3.4. H_2_S Adsorption

The effects of inlet concentration were measured in samples with H_2_S concentrations of 150 ppm, 300 ppm, and 500 ppm at ambient temperature. The breakthrough curves for 13X and modified zeolites are demonstrated in Figure 4. The concentration of H_2_S in the outlet stream was zero for a significant amount of time before it broke through. The experiments were stopped when the outlet concentration reached 10% of the initial concentration, indicating an effective adsorption time [27]. Adsorption capacity of the 13X, AgI-13X, AgII-13X, and AgIII-13X was calculated from the experimental breakthrough curves and is shown in Table 3. Breakthrough time was determined when the outlet concentration was 1 ppm. Experiments were run until the outlet concentration reached 10% of its initial concentration to show how the breakthrough curve evolved after breakthrough time. Increased inlet H_2_S concentration resulted in earlier breakthrough times for all samples, as anticipated. The 13X molecular sieve exhibited the earliest breakthrough time over all concentration ranges (150–500). It is noteworthy to mention that the breakthrough time for AgIII-13X was detected earlier than that of AgII-13X. However, the breakthrough curves demonstrated that AgIII-13X was capable of adsorbing substantial quantities of H_2_S molecules even after breakthrough time. Across all ranges, the longest breakthrough time for AgII-13X samples was discovered, indicating their high effective adsorption capacity.

### 3.5. Effect of Inlet Gas Composition

The influence of H_2_S concentration in the inlet gas composition was evaluated to determine its impact on adsorption capacity. Initially, three concentrations of H_2_S in natural gas were considered: 150 ppm, 300 ppm, and 500 ppm. The breakthrough curve, the corresponding breakthrough time, and the effective H_2_S adsorption capacity are shown in Figure 4 and Table 3, respectively. When high H_2_S concentrations of 500 ppm were used, the breakthrough was reached, as expected, significantly earlier (Figure 4c) than with the concentration of 150 ppm. At a 300 ppm H_2_S inlet concentration, a greater adsorption capacity of 13.05 mg/g was achieved for AgII-13X. The lowest adsorption capacity of 0.238 mg/g was observed for non-modified 13X when the H_2_S inlet concentration was 150 ppm. Table 4 provides the breakthrough times for the samples.

### 3.6. Effect of Ag Concentration

The effect of the concentration of Ag ions on the adsorption capacity of modified zeolite during H_2_S uptake was investigated in the range of molar concentrations of 0.02–0.1M AgNO_3_ water solution. The results are reported in Figure 4 and Table 3, respectively. It can be seen in Figure 4 that the increase of silver ions on zeolite samples led to an increase in breakthrough time and H_2_S adsorption capacity. However, when the silver ion concentration was too large (Figure 4a for AgIII-13X, 0.1 M AgNO_3_), the breakthrough time was observed earlier with respect to AgII-13X, resulting in a decrease in the adsorption performance. In spite of observing an earlier breakthrough time for AgIII-13X, its breakthrough curve changed marginally compared to those of the other samples. The use of a high-AgNO_3_ solution concentration resulted in an increase in the cost of the adsorbent. Therefore, modification of the 13X molecular sieve using 0.05 M AgNO_3_ solution was believed appropriate. It was determined that AgII-13X possessed a greater adsorption capacity, measuring 13.06 mg/g. AgII-13X showed about 50 times more adsorption capacity than non-modified 13X, which only had 0.238 mg/g of adsorption capacity.

### 3.7. Adsorption Mechanism

After the procedure was started, it was observed that the initially white adsorbent surface underwent a color change, transitioning to a darker shade. This alteration in coloration served as an indication that chemical adsorption was taking place. Sodium (Na^+^) ions present in 13X replaced silver (Ag^+^) cations, leading to a subsequent chemical reaction with hydrogen sulfide (H_2_S) molecules, resulting in the formation of black silver sulfide (Ag_2_S). The possible adsorption mechanism of H_2_S on silver-modified 13X was presented in Figure 5. Na^+^ cations present in 13X molecular sieves can be replaced by Ag^+^ cations. The H_2_S molecules are strongly attracted to the Ag^+^ cations. In fact, the large number of Ag cations contained in the zeolite had to increase their attraction to H_2_S molecules. However, the zeolite interacts with H_2_S molecules at the initial time of adsorption of Ag ions on the surface of the zeolite, forming Ag_2_S and preventing other Ag ions in the inner layer from showing activity. As a result, no matter how much the amount of Ag cations contained in the zeolite increases, it does not have a positive effect on the increase in H_2_S adsorption. Although AgII-13X contains fewer silver cations than AgIII-13X, its adsorption capacity becomes higher. π-complexation and sulfur-metal (S–M) bond formation may take place between sulfur compounds and metal ions. Previous research also stated that [24] the S–M bond was found to exist between the metal ion and H_2_S. The Ag–sulfide bond was found to have the highest strength according to the Mayer bond order (0.639), which was determined by employing density functional theory (DFT).

### 3.8. Adsorption Isotherms

The Langmuir and Freundlich adsorption models were applied to assess adsorption parameters and to investigate adsorption mechanisms at ambient temperature. The Langmuir model describes monolayer adsorption of adsorbate onto homogenous solid surface sites, while the Freundlich model does not have a maximum adsorption limit. Two adsorption models were implemented for AgII-13X since it showed a high adsorption capacity (Figure 6).

The following Langmuir isotherm equation was used:(2)$$q_{e}=\frac{K_{L}q_{max}C_{e}}{1+C_{e}K_{L}}\;$$
where q_e_ and C_e_ are the H_2_S uptake and equilibrium concentration, respectively, K_L_ is the Langmuir isotherm constant related to the binding energy, and q_max_ is the theoretically calculated adsorption capacity.

The Freundlich model adsorption parameters were obtained using the following Equation (3):(3)$$q_{e}=K_{F}C_{e}^{1/n}$$
where K_F_ is a Freundlich constant or maximum adsorption capacity, *C_e_* is the concentration of adsorbate under equilibrium condition (mg/L), *q_e_* is the amount of adsorbate adsorbed per unit mass of adsorbent (mg/g) giving the value indicating the degree of linearity between the adsorbate solution and the adsorption process The adsorption isotherm parameters for the Langmuir and Freundlich models were provided in Table 5.

The average determination coefficient (R^2^) for AgII-13X zeolites was 0.9346 in Langmuir and 0.9006 in Freundlich, indicating that Langmuir’s isothermal model was better in our case. The maximum adsorption capacity was calculated as 29.42 mg/g, which was higher than the effective adsorption capacity of 13.06 mg/g.

## 4. Conclusions

Several promising results have been reported on the modification of the commercially available 13X molecular sieve adsorbent with silver ions and its application in the separation of H_2_S from various mixtures. However, there is no comprehensive study in the available literature on the extraction of H_2_S from natural gas using Ag-modified adsorbents. In this work, 13X molecular sieve was modified with silver to purify natural gas from H_2_S. Various concentrations of AgNO_3_ solution were used to modify 13X to assess the effect of the Ag^+^ cation level on the adsorption properties. The effect of the inlet concentration was evaluated in the range of 150–500 ppm. The results showed that the ion exchange of 13X molecular sieve with silver ions had a positive effect on the increase in adsorbent capacity. The highest adsorption capacity of 13.05 mg/g was reached using AgII-13X zeolite and was highly effective in removing H_2_S from natural gas. Silver-modified 13X molecular sieve is a promising material that can be utilized as an adsorbent for the purification of various gases from H_2_S, including natural gas, refinery gas, coal gas, syngas, and biogas. The findings of this study have to be seen in light of some limitations. First, regeneration of the spent adsorbent was not investigated due to lack of regeneration equipment. Second, the effect of other natural gas components, such as carbon dioxide, nitrogen, and other sulfur components, was not evaluated. This will be crucial in future attempts to explore the capabilities of silver-modified 13X molecular sieve under varied conditions.

## Figures and Tables

**Figure 1 materials-17-00165-f001:**
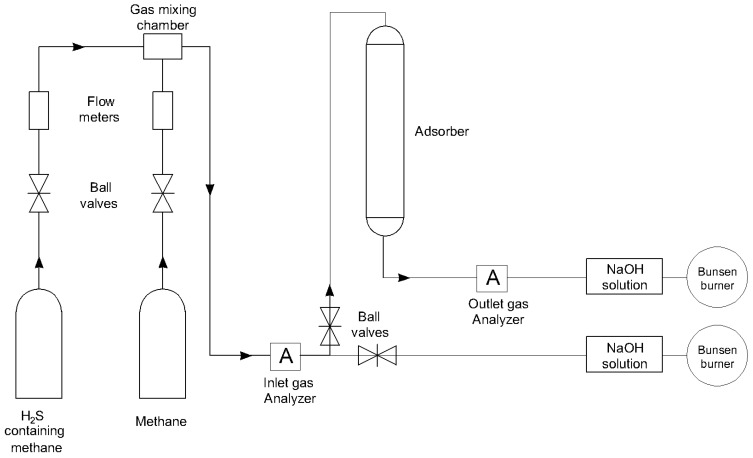
Laboratory setup for H_2_S adsorption.

**Figure 2 materials-17-00165-f002:**
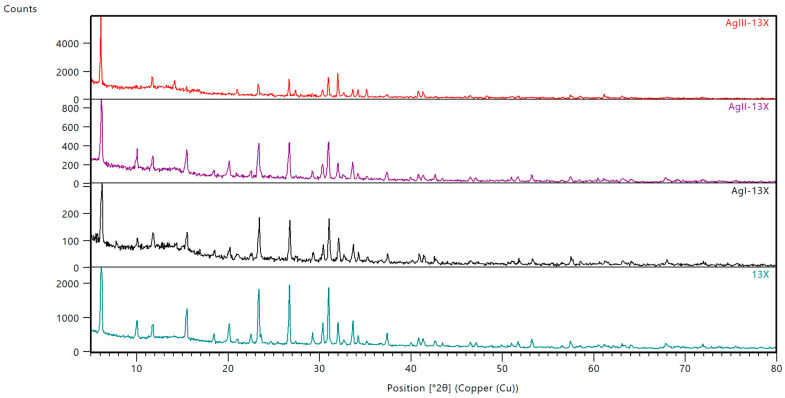
XRD pattern registered for the 13X, AgI-13X, AgII-13X, and AgIII-13X samples.

**Figure 3 materials-17-00165-f003:**
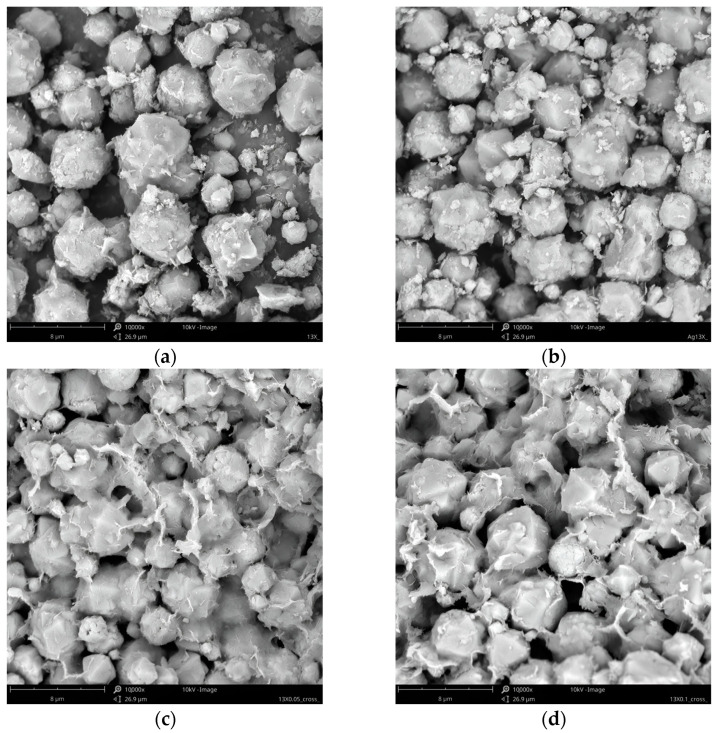
SEM images of (**a**) 13X; (**b**) AgI-13X; (**c**) AgII-13X; and (**d**) AgIII-13X, mag. 10,000×.

**Figure 4 materials-17-00165-f004:**
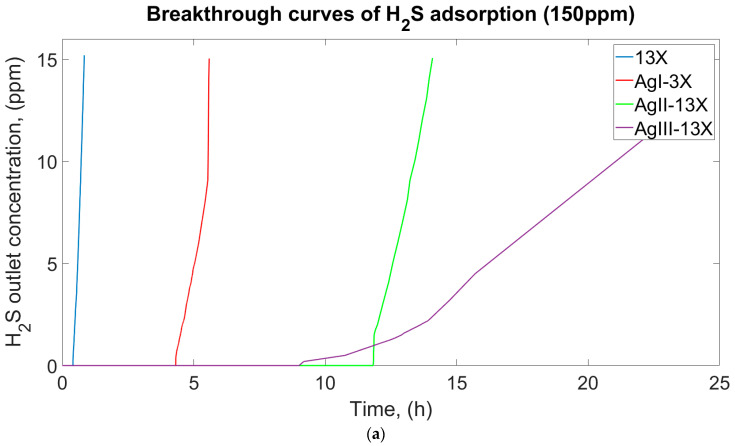
Breakthrough curves of zeolites in the range of (**a**) 150 ppm, (**b**) 300 ppm, and (**c**) 500 ppm H_2_S in methane.

**Figure 5 materials-17-00165-f005:**
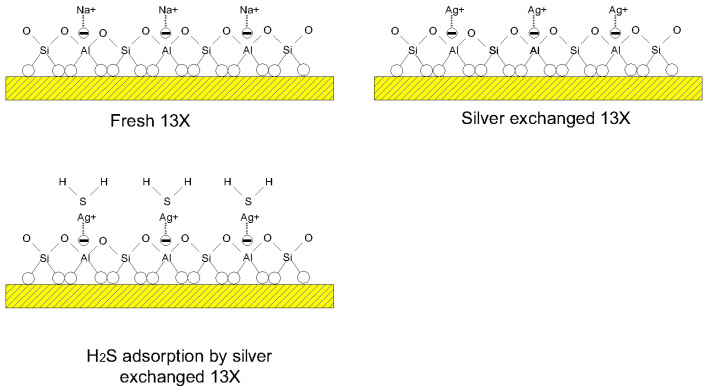
The possible adsorption mechanism of H_2_S on silver-modified 13X.

**Figure 6 materials-17-00165-f006:**
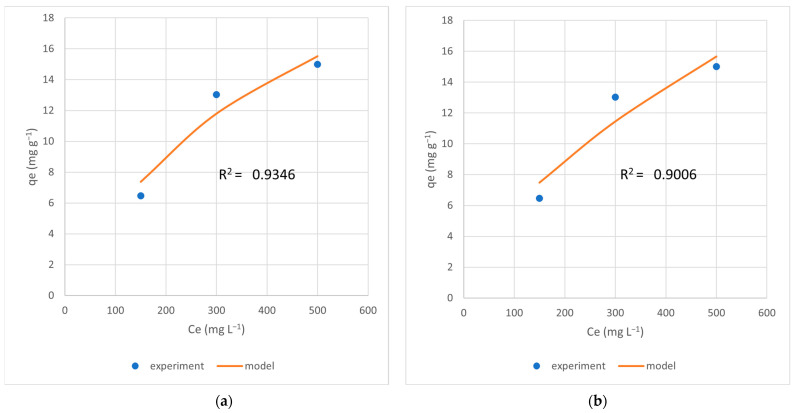
Langmuir (**a**) and Freundlich (**b**) adsorption isotherm models at ambient temperature for AgII-13X.

**Table 1 materials-17-00165-t001:** The element content in 13X, AgI-13X, AgII-13X, and AgIII-13X was determined by the EDS analysis.

Element	Elemental Composition in wt.%			
	13X	AgI-13X	AgII-13X	AgIII-13X
O	55.81	51.83	46.80	39.24
Si	21.20	18.93	16.20	14.31
Al	15.74	12.27	9.40	11.13
Na	6.89	5.79	2.38	2.07
Mg	0.35	0.94	1.55	0.43
Ag	-	10.24	21.32	32.38
Σ	99.99	100	97.65	99.56

**Table 2 materials-17-00165-t002:** BET surface area of the samples.

Adsorbents	BET Surface Area (m^2^/g)
13X	501.33
AgI-13X	436
AgII-13X	416
AgIII-13X	405

**Table 3 materials-17-00165-t003:** Adsorption capacity of the samples.

Adsorbents	Effective Adsorption Capacity (mg/g)		
	150 ppm	300 ppm	500 ppm
13X	0.238	0.254	0.26
AgI-13X	2.405	7.92	8.44
AgII-13X	6.47	13.05	11.44
AgIII-13X	4.92	9.1	9.15

**Table 4 materials-17-00165-t004:** Breakthrough times for the 13X, AgI-13X, AgII-13X, and AgIII-13X samples in the range of 150–500ppm.

Adsorbents	Breakthrough Time (h)		
	150 ppm	300 ppm	500 ppm
13X	0.435833	0.232778	0.1425
AgI-13X	4.403889	7.253611	4.638611
AgII-13X	11.84806	11.9475	6.284444
AgIII-13X	11.87917	8.338889	5.03

**Table 5 materials-17-00165-t005:** Adsorption isotherm parameters for AgII13X.

Langmuir	Value	Freundlich	Value
***K_L_* (µM^−1^)**	0.002229	***K_F_* (mmol m^−2^ µM^−1^/n)**	0.347795
***R*^2^ (*C_e_* vs. *q_e_*)**	0.9346	***R*^2^ (*C_e_* vs. *q_e_*)**	0.9006
***q_max_* (mg g^−1^)**	29.42685	** *n* **	1.632228

## Data Availability

Data are contained within the article.

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
