# Peer review of "Hydrogen Sulfide Adsorption from Natural Gas Using Silver-Modified 13X Molecular Sieve"

_materials, 2023, doi:10.3390/ma17010165_

Round 1

Reviewer 1 Report

Comments and Suggestions for Authors

The work shows the preparation and characterization of Ag-modified 13X molecular sieve adsorbent. H2S adsorbent systems were prepared with three levels of silver concentration. The materials were characterized using XRD, SEM, and BET techniques.

The topic is of high interest in the field of extraction and purification of gas streams. The corrosion generated by H2S in distillery units can be minimized through the use of efficient adsorbents. 

The manuscript is short and well written, but before being accepted it is suggested to review some points detailed below:

Regarding the silver species:

What is the oxidation state of the silver species supported on the molecular sieve? XPS?

Are the authors sure that only exchanged silver species exist? Why do the authors rule out the presence of segregated species?

The concentration of silver ions present in the adsorbents is not reported. The concentration shown by the SEM-EDS technique may be higher than the nominal one.   I suggest determining the concentration of silver in the initial silver solution and the solution after the exchange. This procedure can be done with simple volumetric techniques because the silver concentration is high (0.02 M-0.1M).

10-30 wt.% are high concentrations of silver. The authors should critically compare the performance of the materials studied in this work with those reported in the literature.

The novelty of this work must be highlighted.

Reviewer 2 Report

Comments and Suggestions for Authors

Interesting manuscript on the adsorption of hydrogen sulfide (H2S) from natural gas using Ag-13X molecular sieve. The manuscript evaluates the adsorption capacity of Ag-modified 13X molecular sieves with unmodified 13X molecular sieves. The data provided is good, however, it needs to be slightly revised before this manuscript can become suitable for publication. My questions/suggestions are as follows:

My questions/suggestions:

1. To remove residual gases and traces of humidity, the authors heated the adsorbents in an oven at 110 °C overnight. Were these conditions sufficient? Typically, zeolites are degassed at high temperatures ~ 300 °C and under vacuum.

2.         The authors can normalize the XRD curves for the samples. Currently, it is difficult to clearly see curves for the two Ag(I&II)-13X samples.

3. In Lines 255-256, the authors say that excessive metal ions might clog zeolites’ pores and prevent H2S from adsorbing on AgIII-13X samples. However, the specific surface area (Table 2) did not change significantly from AgII-13X to AgIII-13X. So, the explanation may not be completely true. Have the authors measured the pore volume for all these materials?

4.         In the conclusions section, authors say AgII-13X zeolite is highly effective at removing H2S from a variety of gases that contain methane. However, here authors have only tested methane 2.5 and methane with 5000 ppm H2S. Please correct it.

5.         The authors can probably include H2S adsorption data for 13X from other literature for the sake of comparison. There are a couple of reports that have studied Cu-13X as well.

6.         In the introduction, some other literature reports can be included:

a.     https://doi.org/10.1016/j.seppur.2022.121448

b.     https://doi.org/10.1016/j.envpol.2022.120219

Reviewer 3 Report

Comments and Suggestions for Authors

Title: Hydrogen sulfide (H2S) adsorption from natural gas using silver-modified 13X molecular sieve

Dear Authors,

I have carefully reviewed the manuscript "Hydrogen sulfide (H2S) adsorption from natural gas using silver-modified 13X molecular sieve" and have several comments and suggestions for improvement. 

Title:

  • Please remove "H2S" from the title to enhance clarity and conciseness.

Affiliations:

  • The affiliations of the authors are not provided. Kindly ensure that this information is included.

AgII-13X:

  • Please explain what "AgII-13X" refers to in the text.

Abbreviations in Abstract:

  • Avoid abbreviations in the abstract for improved readability.

Abstract:

  • Expand the abstract by including more words highlighting the research's novelty and practical utility.

Introduction:

  • Lines 32-33: The statement "The process of absorption involves a chemical reaction between acid gases and solvents; hence, it is referred to as chemisorption" requires clarification. Please review and rephrase if necessary.
  • Suggest discussing H2S absorption using different absorbents and identifying the most effective method for H2S removal, either through absorption or adsorption.
  • Please check the reference citation format for this journal.

Materials and Methods:

  • Improve the characterization section by providing analysis conditions and instrument names and models.
  • Modify lines 116-118 for clarity.
  • Include the pH value of the NaOH solution used in line 123.
  • Clarify whether separate CH4 and H2S gases were purchased for the experiment or if a mixture of 5000 ppm H2S in CH4 was used. Provide details on how different ppm gas mixtures were prepared.

Results and Discussion:

  • Include the XRD spectrum of AgIII-13X in Figure 2.
  • Explain why breakthrough curves in Figure 4 differ from the 'S' shape.
  • Elaborate on the mechanism section with graphical presentations and clarify the superior performance of AgII-13X compared to other substances, even with differences in surface area values.
  • Ensure all figures and text within figures are presented in high resolution.

Conclusions:

  • Revise the conclusion to include the practical utility, potential for future research, and study limitations.

Round 2

Reviewer 1 Report

Comments and Suggestions for Authors

Dear authors,

I have carefully read the new version of the work and your responses to the first revision. I consider that my main doubts/questions have not been answered.

I consider that the work is good, but I would like some points to be explained in greater detail.

It is necessary to clarify the nature of the silver phases supported/exchanged in the molecular sieve. For this, it is necessary to know the concentration of silver supported. Please determine or report the exchange capacity of the molecular sieve. A silver concentration of 30 wt.% is high, that amount of silver ions can be exchanged on the sieve or can also remain as Ag2O on the surface. It is worth mentioning that the materials were treated at 600°C.

Again, I repeat that the determination of the amount of silver impregnated/exchanged on the sieve can be easily determined.

Please review the XRD analysis, the diagram of the AgIII/13X material differs from those obtained with materials with lower silver content (for example, 2 tetha=29-35 °)

I repeat that it is of interest to compare the results obtained with these materials with other materials based on the use of silver (not necessarily supported in 13X molecular sieve)

Author Response

We appreciate your assistance in revising our manuscript. It contributed to its increased value.

Kindly refer to the file attached for the responses.

Reviewer 3 Report

Comments and Suggestions for Authors

The authors have made improvements to the manuscript and have addressed the previous comments by providing more clarity. The manuscript could be acceptable for publication. 

Author Response

We express our gratitude for the revision you have provided. The revision contributed to the enhancement of our paper intended for publication.

Round 3

Reviewer 1 Report

Comments and Suggestions for Authors

The authors modified the manuscript. I suggest accepting this new version.